# Are the shoulder joint function, stability, and mobility tests predictive of handstand execution?

**Roman Malíř** [ORCID]*, **Jan Chrudimský, Adam Provazník, Vít Třebický**

Faculty of Physical Education and Sport, Charles University, Prague, Czech Republic

* malir@ftvs.cuni.cz

## Abstract

Handstand is a basic element common across gymnastic disciplines and physical education classes that is frequently evaluated for quality in competition or skill acquisition. The correct handstand execution relies on maintaining balance, for which the shoulders seem particularly important. This study explores the relationship between shoulder joint function and the quality of handstand execution in novice college athletes (n = 111; aged 19–23 years). We assessed the shoulder joint function using standardized field tests (Upper Quarter Y Balance Test and Closed Kinetic Chain Upper Extremity Stability Test) and evaluated handstand execution on official rating scale. Ordinal logistic regression models showed no relationship between the quality of handstand execution (E-score) and measures of shoulder joint stability or mobility in our sample (POR = 0.97 [0.91, 1.03] and 1.00 [0.91, 1.09] for E-score). Two major factors may have caused an observed pattern of results. Firstly, the standardized tests assess shoulder joints in different loads and ranges of motion compared to handstands. Secondly, our novice sample was not able to perform the handstand sufficiently well. In our sample of novice college athletes, shoulder function seems not related to handstand execution as other latent factors hindered their performance.

**Data Availability Statement:** All necessary files included in available Supplmental online material.

**Funding:** This research was supported by the Cooperatio Programme, research area Sport Sciences – Biomedical & Rehabilitation Medicine

## Introduction

Handstand is an essential and frequent element in gymnastics and physical education (PE). It is a fundamental skill [1–4] performed both in dynamic (performed as position passing through swing exercises i.e., parallel bar, pommel horse) and static (maintaining a balanced inverted body position, i.e. floor exercises, rings) forms. The static form of the handstand is of particular relevance as it is frequently the initial and/or the final position of many figures [5, 6].

Apart from gymnastics, it is a standard skill for assessing movement literacy [7]. The capability and proficiency of performing a handstand are crucial for learning more advanced and combined elements, such as handstand to forward roll, backward roll to handstand, or handsprings [4, 8]. These elements are commonly used in physical education [8] from elementary to high school levels and are frequently researched among college physical education (PE)

(SPOB). The funders had no role in study design, data collection and analysis, decision to publish, or preparation of the manuscript.

**Competing interests:** The authors declared no potential conflicts of interest concerning the research, authorship, and/or publication of this article.

students [8–10]. As gymnastics is commonly included in the PE curriculum of primary and secondary education [11–14] across many educational systems, prospective PE teachers are supposed to be familiar with fundamental gymnastic element such as rolling, hanging, swinging, or supporting [11, 15], among which the handstand should be included [11]. Apart from understanding the gymnastic element itself, it is desirable for PE teachers to be able to demonstrate the particular element [15], as observing a performed element before learning it increased skill acquisition in students [16].

Proper static handstand is characterized as a maintained balance in an inverted straight body position [17, 18] with hands in contact with the ground or support surface [18]. During such handstand, arms should be in ~180˚ flexion [4] with extended elbows. From the perspective of performance evaluation, static positions such as a handstand, must be held for minimal requested duration of two seconds, with shorter durations being penalized [19] and balancing a handstand with larger corrective movements and sway also results in a worse performance rating (e.g., a higher score deductions for the execution [20]). Maintaining the static handstand is a complex interplay of various factors [21–25] mostly affected by the reciprocal coordination of the wrists, elbows, shoulders, and hips [1, 5, 26–29]. The wrists and wrists' torque are considered as the most important for balance maintenance [1, 5, 27], followed by flexion in elbows [28] and hips [1, 5, 26, 29] allowing for corrective movements. Some authors [4, 27, 29, 30] have suggested that for performing a handstand, shoulders are an important joint group as their two main capacities—mobility and stability [31] are influential in the center of mass (COM) shifting [1]. Therefore, it seems that the shoulder joint function may play a substantial role in static handstand execution quality. Yet, the role of shoulders for maintaining a stable position in a handstand is less recognized [1, 21].

This study therefore aims to investigate the relationship between standardized shoulder joint mobility and stability tests and the quality of handstand execution among prospective PE teachers. We expect that the quality of handstand execution is influenced by the shoulder mobility and stability function and that the results of standardized shoulder joint tests can predict the quality of handstand execution assessed by the gymnastics scoring system. We thus predict that participants with better scores in shoulder joint stability and range of motion tests will also achieve better handstand quality scores.

## Materials and methods

The study took part during the winter and summer terms in 2021 and was conducted in the sports gym of the Faculty of Physical Education and Sport of Charles University. All procedures were carried out in accordance with the Declaration of Helsinki and under relevant safety rules regarding the COVID-19 pandemic. The Institutional Review Board of the Faculty of Physical Education and Sport of Charles University approved the study (198/2020). All participants were informed about the study goals and signed a written informed consent before participation.

### Participants

We recruited a convenience sample [32] of all first-year bachelor's degree students available at the Faculty of Physical Education and Sport at Charles University from Physical Education and Sport and Coaching study programmes. The reached sample consisted of 111 students (35 women and 76 men) aged 19–23 years (mean = 20.21, SD = 1.02 years) (further sample descriptives are available in the Supplemental digital content). All participants were active athletes, who passed a semester-long "Basics of gymnastics" course focused on the fundamentals of gymnastics, including learning handstand with all necessary drills and preparatory exercises

used in learning static handstand. The Basic gymnastics course curriculum does not differ between the two study programmes. In addition, before the study onset, all participants underwent two 45minute lessons directly focused on static handstands performed. On this basis, we assumed that students are able to master handstand to a sufficient degree. Only participants with no history of shoulder surgeries or acute upper limb injury were allowed to participate.

## Data collection

**Procedures.**   Data collection took place during the last lesson of the Basic gymnastics course at the end of the winter and summer terms. The participants were divided into two groups based on the different study programs. First group of 55 participants were tested at the end of the summer term on 24[th] to 28[th] of May (students of Physical Education and Sport study program), and the second group of 56 at the end of the winter term on 6[th] to 10[th] of December (students of Coaching study program) in 2021.

Participants were familiarized with the course of the study and all testing procedures. Subsequently, they obtained a protocol form that included an assigned ID and contained fields to fill in the results of all testing procedures; each participant carried the protocol form throughout the testing period (see in the Supplemental digital content). Next, all participants underwent anthropometric measurements of body height, weight, and arm's length (used for UQYBT score calculation, see below). Arm length was measured in the upright standing position with arms abducted to 90˚. The lengths of both arms were measured from the C7 vertebra spinous process to the dactylion as per [33]. Subsequently, the participants performed a standardized and supervised (by two members of the authors collective) 8-minute gymnastic warm-up (mobilization and stretching) predominantly focused on the upper body and shoulder joints. After the warm-up, participants were randomly divided into groups, which were randomly assigned to test stations. After finishing the task at a given station, the group continued to the next (randomized) station with a fixed 5-minute resting period between tests.

Handstand execution was recorded, and the quality was subsequently evaluated using E-score scale. The shoulder joint stability was examined using two standardized field tests: Upper Quarter Y Balance Test (UQYBT) [34] and the Closed Kinetic Chain Upper Extremity Stability Test (CKCUEST) [35].

**Station 1—Handstand execution, recording, and evaluation.**   Each participant started from the middle of a firm 5 cm thick mat (200×100 cm). A soft 10 cm thick mat (200×100 cm) was placed in front of the firm mat as a safety precaution in case of a fall. There was approximately a 40 cm wide gap between the two mats where participants were to put their hands during handstand execution. Participants were instructed to perform a handstand and keep the balance for 2 seconds without additional movements. Each participant had the option to choose a starting position from two predefined (1) starting from a front support position with hands put on the ground and one leg bent 2) starting from a standing position with arms up dynamically transferring into directly putting hands in the gap) and execute the handstand directly from this position. The research assistant gave verbal instructions "start" to start and "stop" to finish the handstand attempt. Each participant had a maximum of three attempts to perform the handstand. The first successful attempt was recorded.

Two digital cameras were used to record the execution of the handstands of every participant. The first (front view) camera (DSLR Canon EOS 550D equipped with Canon Zoom Lens EF-S 18-135mm 1:3.5–5.6 IS set to its widest setting, recording in 1080p, 30fps) in landscape orientation was placed on a tripod approximately 1 meter above the floor and 6 meters in front of the participant. The second (side view) camera (Canon HF-R17 with Lens 3.0–60.0mm 1:1.8 set to its widest setting, recording 1080p and 25fps) was also placed on a tripod

approximately one meter above the floor and 5 meters away from the right side of the participant during handstand execution. The recording of each participant included the starting position and the successful attempt; all body segments were always visible during the handstand, except for the ankles and feet, which were irrelevant for later evaluation.

**Handstand quality evaluation.** As in previous study [8] that assessed handstand execution in physical education classes, we rated the quality of the handstand using the E-score evaluation of errors in *execution and technical performance aspects* as defined by men's artistic gymnastics code of points (MAG CoP) [20]. E-score starts at 0 points for flawless execution and adds points or their fractions for each error/ deviation in execution. In our study, for any deviation of the angle in the hips, knees, shoulders, and elbows from the correct position, up to 0.5 points were added (0.1 error points = up to 15˚; 0.3 error points = 16˚ - 30˚; 0.5 error points = more than 30˚). The addition of 0.3 points was accounted for when participants kept their legs apart during the handstand position. When participants were unable to hold the position for the full 2 seconds, 0.3 points were added, and 0.5 error points if there was no holding of the position during execution. When a participant fell from the handstand (uncontrolled descent from the handstand position), 1 point was added.

Three members of the authors' collective, each with more than ten years of practical experience in artistic gymnastics, independently assessed each participant's performance of a handstand by observation of the recorded frontal and side view.

**Station 2 –The Upper Quarter Y Balance Test.** For the UQYBT, we used the standardized procedure for the Y balance test kit [33]. The testing position was a single arm front support with legs a pelvic width apart, keeping a straight body position. The hand of the support arm was positioned next to the red line markings on the middle block of the test kit [34]. Participants were instructed to use their free hand to move sliding blocks along three axes (mediolateral, inferolateral, and superolateral) as far as possible. Bending the elbow of the support arm, disrupting the prescribed body position, or touching the ground with a free hand was not allowed. Each participant had three attempts for each arm, with the right arm first tested (failed attempts were not counted). We set the breaks between attempts to 1 minute. Following the UQYBT protocol [33], we computed the score for the right and left arms separately as a sum of the furthest reaches (cm) in all three axes divided by the corresponding arm length multiplied by three and then multiplied by 100:

$$\text{UQYBT Right} = 100\left(\frac{\sum(\max med + \max \inf + \max sup)}{3(arm\ length)}\right)$$

Note: formula example for UQYBT Right; med = mediolateral directions; inf = inferolateral direction; sup = superolateral direction.

We recorded the UQYBT score for the right arm (UQYBT Right), left arm (UQYBT Left) and the total score (UQYBT Total). The UQYBT Total score was obtained as a mean of UQYBT Right and UQYBT Left. Only the UQYBT Total score was used for subsequent analyses.

**Station 3 –The Closed Kinetic Chain Upper Extremity Stability Test.** The CKCUEST test was performed in a wide push-up position with hands 1 yard (91.5 cm) apart with a straight body position and legs a pelvic width apart [36]. From this position, participants were instructed to lean over one hand (supported hand), touch the dorsum of the supported hand with the free hand, return the free hand to starting position, and repeat the task with the other hand. The main goal of this task is to perform hand touches as fast as possible, regardless of which limb starts. Each participant had three 15 second attempts [35] with 1-minute breaks

between attempts (failed attempts were not counted). Each participant began and finished the test with verbal cues "start" and "stop" from the research assistant. For the CKCUEST score, we calculated the mean number of hand touches for all three trials of the CKCUEST [35] and used it for the subsequent data analysis.

### Data processing and statistical analysis

All data were entered into MS Excel 2016 spreadsheets, subsequently processed and analyzed using R version 4.3.2 [37] via RStudio 2023.06.0 IDE [38].

**Concordance between evaluators.**   Kendall's coefficient of concordance from the *rcompanion* package [39] was used to assess the concordance between three E-score evaluators. We selected W ≥ 0.7 (p ≤ 0.05) as a sufficient level of concordance. The final E-score was based on the mean of the assigned ratings of all three evaluators. Fig 1 highlights the frequency of reached E-score.

**Exploratory data analysis.**   We assessed the normality distribution of all continuous variables (UQYBT Right, UQYBT Left, UQYBT Total, CKCUEST & E-score) using Shapiro-Wilk test. Next, we assessed the equality of variance between all continuous variables included in data analysis (UQYBT Total, CKCUEST, E-score) by Fligner-Killeen's test using *fligner.test* function from *stats* package [37].

Using the *cor.test* function from *stats* package [37], we explored associations (and possible collinearities) between variables using Pearson's r with its 95% CI (for parametrically distributed variables, i.e., UQYBT Right and UQYBT Left, UQYBT Total and CKCUEST). The threshold of close association and interchangeability was set to ≥ 0.7. If the association between a pair of variables would have reached this predefined value, we would use only one of them.

**Relationship between quality of handstand and shoulder stability and mobility tests.** We used a regression model to test the relationship between handstand execution quality and shoulder stability and mobility tests. Variance inflation factor (VIF) in *car* package [40] was used to assess the assumption of multicollinearity between predictors (i.e., UQYBT Total, CKCUEST) with a predefined level of multicollinearity < 5.0 [41]. If the VIF criterion would be greater than the predefined value, we would remove highly correlated predictors from the models [41] to avoid increasing standard errors estimates of coefficients [41, 42].

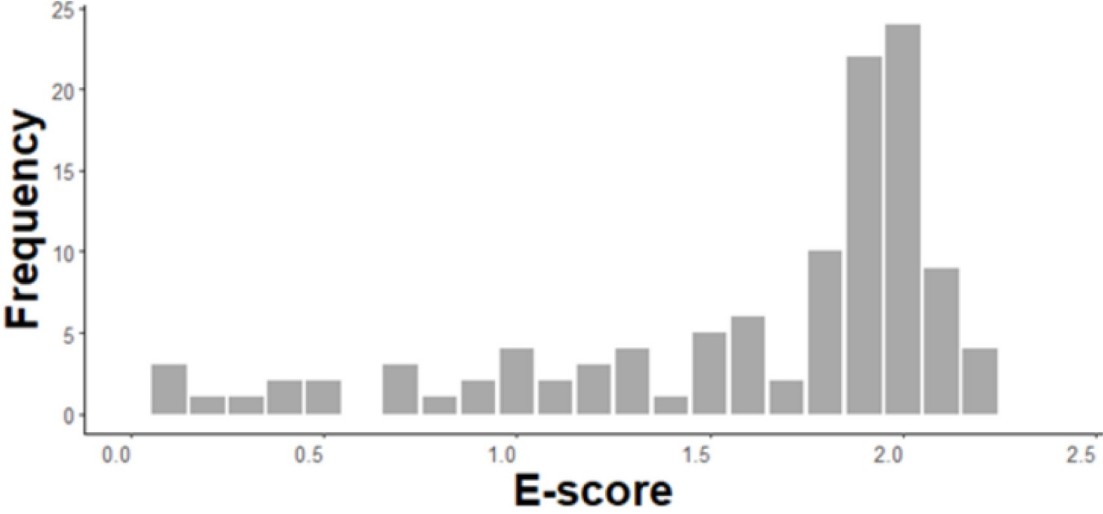

**Fig 1. Frequency of reached E-score.**

We set up ordinal logistic regression model to analyze the relationship between the quality of handstand execution (E-score) and the results of the stability, functionality, and mobility of the shoulder joint tests (UQYBT Total, CKCUEST). Due to the ordinal scaling of the E-score, we fitted an ordinal logistic regression (*formula*: E-score ~ UQYBT Total + CKCUEST) using *MASS* package [43]. We used *performance* package [44] for $R_{McFadden}^2$ and $R_{McFadden\ adj.}^2$ computations. Anova function from *car* package [40] was used for the computation of $\chi^2$ for all three predictors. Subsequently, we used Brant's test to assess parallel regression assumption (PRA) within the ordinal logistic regression model using *brant* package [45] with a predefined alpha level for PRA of $p \geq 0.05$. The main output of the ordinal logistic regression is reported as proportional odds ratios (POR) for individual coefficients of the model (independent variables) and their 95% CI.

**Supplemental online material.** The dataset file (in.xlsx), commented R script with outputs of detailed results of all performed analyses, and supplementary data analyses are available in the Supplemental digital content of this article.

## Results

Table 1 shows summary statistic for individual test of the shoulder joint function (UQYBT Right, UQYBT Left, UQYBT Total, CKCUEST) and for individual handstand quality scores (E-score).

### Concordance between evaluators

The results of Kendall's coefficient of concordance (corrected for ties) showed sufficient agreement between the three judges for E-score (W = 0.84 [0.84, 1.00], p < 0.001). Therefore, we used mean values of the E-score for each participant in subsequent analyses.

### Data assumptions and exploratory data analysis

According to the results of the Shapiro-Wilk test, all continuous variables met the assumption of normal distribution (all Ws $\geq$ 99, ps $\leq$ 0.94), except the E-score (W = 0.81, p < 0.001). The homogeneity of variance assumption was met between all sets of variables (i.e., UQYBT Total and E-score; CKCUEST and E-score).

Results of UQYBT Right and UQYBT Left were strongly, positively and statistically significantly correlated ($r_{111}$ = 0.78 [0.70, 0.84], p < 0.001). We found a weak negative and statistically non-significant correlation between UQYBT Total and CKCUEST ($r_{111}$ = -0.05 [-0.24, 0.14], p = 0.583).

**Table 1. Sample descriptive statistics.**

| Variable | Mean | Median | SD | Min | Max |
|---|---|---|---|---|---|
| UQYBT Right | 86.06 | 86.5 | 5.77 | 70.2 | 100 |
| UQYBT Left | 85.51 | 85.9 | 5.66 | 73.4 | 100.8 |
| UQYBT Total | 85.78 | 86.5 | 5.4 | 71.8 | 100.4 |
| CKCUEST | 27.84 | 27.84 | 3.77 | 15.67 | 39.33 |
| | **Mean** | **Median (Mode)** | **SD (IQR)** | **Min** | **Max** |
| E-score | 1.62 | 1.9 (2) | 0.54 (0.65) | 0.1 | 2.2 |

Note: UQYBT Right/Left/Total = Upper Quarter Y Balance Test scores for right arm, left arm, and total; CKCUEST = Closed Kinetic Chain Upper Extremity Stability Test Score. Higher E-score is worse score, higher UQYBT and CKCUEST scores are better performances.

**Table 2. Summary of model estimates for UQYBT Total and CKCUEST for E-score.**

| Predictor | Coefficient (β) | Std. Error | t value | p | POR | 95% CI (LL, UL) | $\chi^2$ (p) |
|---|---|---|---|---|---|---|---|
| UQYBT Total | -0.04 | 0.03 | -1.13 | 0.26 | 0.96 | 0.91, 1.03 | 1.28 (0.26) |
| CKCUEST | 0 | 0.05 | -0.01 | 0.99 | 1 | 0.92, 1.09 | 0.00 (0.99) |

The results of VIF criterion analysis regarding the multicollinearity of predictors for each subsequent regression model suggest no multicollinearity (UQYBT Total = 1.00; CKCUEST = 1.00).

### Relationship between quality of handstand and stability and mobility tests

The ordinal logistic regression model reached $R_{McFadden}^2 = 0.002$ ($R_{McFadden\ adj.}^2 = -0.001$) with residual deviances 565.57 ($df_{Residual} = 89$). None of our two measures predicted the observed E-score as statistically significant or with substantial odds (Table 2 and Fig 2). The POR shows that for every 1 cm increase in the UQYBT Total, the odds of being better in the E-score decreased on average by 4%. Within the CKCUEST, every additional touch increased lead on average to 0% change in odds of being better in the E-score; in other words, the CKCUEST does not influence observed E-score in our sample.

The Brant's test of the model holds PRA for UQYBT Total ($\chi^2 = 24.02$, df = 19, p = 0.20), CKCUEST ($\chi^2 = 22.93$, df = 19, p = 0.24) and Omnibus ($\chi^2 = 52.35$, df = 38, p = 0.06).

### Discussion

The ability to perform a handstand is an essential element in gymnastic activities [5, 8] included also in PE [8, 9, 11]. Multiple ways of compensating sway (COM shifts) to maintain handstand position have previously been discussed in the literature [1, 3, 5, 21, 29]. However, they were mostly omitting the role of shoulders, while multiple authors argued and provided supportive evidence for shoulder joint function being an important element for handstand

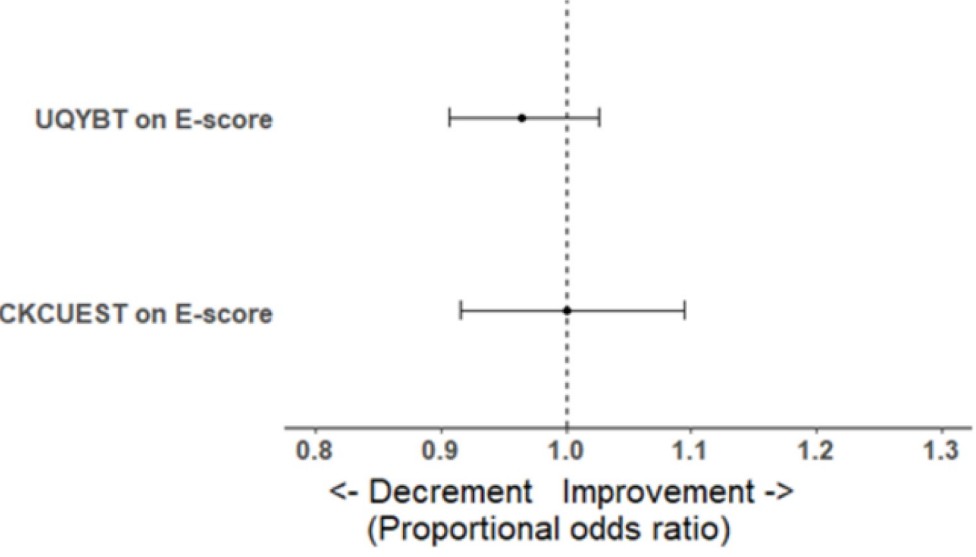

**Fig 2. Proportional odds ratios of shoulder joint function measures on E-score.** Note: Black dots represent observed effect sizes and error bars 95% CIs. The dashed vertical line represents no change in odds. Values below 1 are decrements in odds, and above 1 are improvements in odds.

execution [4, 17, 21, 27, 30]. Therefore, the aim of this study was to investigate the relationship between the quality of handstand execution and shoulder joint function and stability assessed using standardized (UQYBT & CKCUEST) field tests in a sample of prospective PE teachers. Based on our analyses, we observed that the standardized field tests of shoulder joint functioning had no statistically significant effect on E-score rating and, thus, on the quality of handstand execution in our sample.

We used two standardized field tests (UQYBT & CKCUEST) examining aspects of shoulder joint stability and complex function. Although other studies (with smaller samples) report a moderate positive correlation between UQYBT and CKCUEST (e.g., $r_{30} = 0.49$; [46], our results ($r_{111} = -0.05$, [-0.24, 0.14]) are more in accordance with Taylor et al. [47] ($r_{257}$ range = 0.04–0.18), showing virtually no relationship between these two tests, supporting the claim that both tests assess different aspects of shoulder function [47].

In our sample, we observed no changes in the odds of scoring better in the E-score depending on the UQYBT and CKCUEST. This may indicate that the UQYBT and CKCUEST are not necessarily helpful tools for predicting the handstand execution (at least in our sample) and to assess the importance of shoulder in this exercise. The explanation for the negligible odds may lay in the difference of upper extremities loading and position between handstand and selected tests. During handstand execution, the upper extremities are in full flexion (~ 180˚) [48] and therefore, the level of shoulders' aROM should be an important factor contributing to maintaining a handstand. While during UQYBT and CKCUEST, the upper extremities are "only" in the middle flexion (~ 90˚). Though Wattanaprakornkul et al. [49] claimed that similar muscle activity patterns are produced during shoulder flexion torque regardless of the load axis, the difference in range of motion in shoulder flexion may impact the loading of the brachial plexus. As the sternoclavicular joint is involved only in movements of up to ~90˚ of the shoulder flexion [50], the main loading engages rotator cuff of the glenohumeral joint [51], which is critical in a proper shoulder function [52]. The acromioclavicular joint plays a role primarily in scapular elevation [53], which is also a contributor to handstand execution [48, 54]. Thus the differences in aROM can be seen as a considerable factor raised against these field tests of shoulder joint mobility (UQYBT & CKCUEST) in gymnastics Po[19, 55–57].

Further, though the CKCUEST is a standardized test with relatively high reliability, De Oliveira et al. [36] point out the CKCUEST is a discordant due to its systematic error resulting in differences during measurements. During the test, all athletes are instructed to keep their hands at the same distance (36 inches/1 yard/91.5 cm) regardless of maturational or anthropometric characteristics such as shoulder width or arm span, which may systematically affect the results [47].

The level of experience plays a role in the successful handstand execution [23]. We assumed that our sample of college athlete participants preparing to become PE teachers would be able to perform handstands sufficiently well. Based on the performance evaluation, we found that this was not the case; about 64% (N = 71 of 111) of the sample reached higher E-score than 1.7 (see Fig 1) resulting in insufficient handstand quality. This substantially skewed the observed data distribution, not allowing for better estimates of shoulder function on handstand execution. We can only conclude that performing a handstand is a difficult skill for athletes of non-gymnastics backgrounds (e.g., a sample of physical education students from a wide range of different sports backgrounds).

The present study has two major limitations restricting our ability to verify the assumed relationship between the shoulder joint function and quality of handstand execution. The first is related to our sample, where the majority of participants were unable to perform handstands sufficiently well (see Fig 1). Despite of the sample's overall size, only a few participants were able to execute handstand sufficiently. The other limitation is in the different position of the

brachial plexus during mobility and stability testing (using the UQYBT and CKCUEST) and the handstand position. However, we are currently unaware of more suitable test that evaluates shoulder function over a greater ROM of flexion.

To conclude, we observed no association between the shoulder joint function and the quality of handstand execution in our sample. Apart from the potential no *true* effect of the particular shoulder joint functions on handstand execution, these results could be explained by insufficient variability in the handstand capabilities of our sample and different shoulder joint positions during the UQYBT and the CKCUEST compared to those during a handstand. Future research should aim to test the relationship between stability and mobility of the shoulder joint and the quality of handstand execution among experienced gymnasts rather than a heterogeneous sample of PE and sports students.

## Supporting information

**S1 Checklist. STROBE statement—Checklist of items that should be included in reports of *cross-sectional studies*.**
(DOC)

**S1 File.**
(XLSX)

**S2 File.**
(HTML)

**S3 File.**
(R)

## Acknowledgments

The authors are grateful to the collective of assistants who helped with the data collection and all participants for taking part in the study. We thank Bianca Maria Laoëre for English proofreading of the article.

## Author Contributions

**Conceptualization:** Roman Malíř, Jan Chrudimský, Adam Provazník.

**Data curation:** Roman Malíř, Adam Provazník.

**Formal analysis:** Roman Malíř.

**Investigation:** Roman Malíř.

**Methodology:** Roman Malíř, Vít Třebický.

**Project administration:** Roman Malíř.

**Software:** Roman Malíř.

**Supervision:** Vít Třebický.

**Validation:** Roman Malíř, Vít Třebický.

**Visualization:** Roman Malíř.

**Writing – original draft:** Roman Malíř, Adam Provazník.

**Writing – review & editing:** Roman Malíř, Jan Chrudimský, Vít Třebický.

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
