## [Decision Letter · Decision Letter 0]

28 Nov 2023

PONE-D-23-29129Are the shoulder joint function, stability, and mobility tests predictive of handstand execution?PLOS ONE

Dear Dr. Malíř,

Thank you for submitting your manuscript to PLOS ONE. After careful consideration, we feel that it has merit but does not fully meet PLOS ONE’s publication criteria as it currently stands. Therefore, we invite you to submit a revised version of the manuscript that addresses the points raised during the review process.

The reviewers have provided thorough reviews and highlighted shortcomings in the presentation of the work. Please address these point and in particular work on improving the coherence of the manuscript by clarifying the research questions and aims and providing justifications for choices in the methods. 

We look forward to receiving your revised manuscript.

Kind regards,

Aliah Faisal Shaheen

Academic Editor

PLOS ONE

4. We suggest you thoroughly copyedit your manuscript for language usage, spelling, and grammar. If you do not know anyone who can help you do this, you may wish to consider employing a professional scientific editing service.

A clean copy of the edited manuscript (uploaded as the new *manuscript* file).

“This research was supported by the Cooperatio Programme, research area Sport Sciences – Biomedical & Rehabilitation Medicine (SPOB).”

6. Please include a caption for figure 2.

7. We note that Figures 1 and 2 in your submission contain copyrighted images. All PLOS content is published under the Creative Commons Attribution License (CC BY 4.0), which means that the manuscript, images, and Supporting Information files will be freely available online, and any third party is permitted to access, download, copy, distribute, and use these materials in any way, even commercially, with proper attribution. For more information, see our copyright guidelines: http://journals.plos.org/plosone/s/licenses-and-copyright.

1. You may seek permission from the original copyright holder of Figures 1 and 2 to publish the content specifically under the CC BY 4.0 license.

Reviewers' comments:

Reviewer's Responses to Questions

**Comments to the Author**

1. Is the manuscript technically sound, and do the data support the conclusions?

Reviewer #1: Partly

Reviewer #2: Yes

2. Has the statistical analysis been performed appropriately and rigorously? 

Reviewer #1: Yes

Reviewer #2: Yes

3. Have the authors made all data underlying the findings in their manuscript fully available?

Reviewer #1: Yes

Reviewer #2: Yes

4. Is the manuscript presented in an intelligible fashion and written in standard English?

Reviewer #1: Yes

Reviewer #2: Yes

5. Review Comments to the Author

Reviewer #1: Introduction

The introduction is based on citations of old publications, ie. 1, 4, 9, 10, 11, 16, 23, 29, 30, 31, 33, 35. Please rewrite the introduction, and discuss these issues in the introductory part by referring to the following article:

Puszczałowska-Lizis E, Omorczyk J. The level of body balance in standing position and handstand in seniors athletes practicing artistic gymnastics. Acta of Bioengineering and Biomechanics 2019; 21 (2): 37-44. DOI: 10.5277/ABB-01352-2019-02.

The study lacks of a clear aim and research questions.

Material and Methods

Please provide the sample size calculation.

What was the method of random selection? Or was this a convenience sample?

A clarification of the selection criteria would be required.

Results

Medians should be provided in the tables.

Discussion

In the Discussion, the authors present the results in the form of numerical data, which should not be the case. The Discussion is rather superficial and did not offer sufficient explanations for the mechanistic reasons behind the findings.

You should add some limitations of the study.

Conclusions

The work lacks clearly formulated conclusions. Please specify the conclusions in points as answers to the research questions.

References

The reference list contains too many old positions, ie. 1, 4, 9, 10, 11, 16, 23, 29, 30, 31, 33, 35, 41. Please edit.

References are quoted carelessly, with errors - please correct them.

General comments to the Authors

That said, my comments are offered with the intent of helping the authors improve this manuscript. When the authors address these issues I will be able to comment definitively and make the final decision.

Reviewer #2: Presented study aimed to examine the relationship between shoulder joint mobility and stability and the quality of handstand execution among prospective PE teachers. While the manuscript has potential and deal with interesting issue, some substantial changes should be considered before publication. Most crucial, that there was about 60% participants that were unable to handstand and thus it could disturb the analysis outcome, especially that the research method (AQV and E-score) was not sensitive enough (only 4 or 5 levels?).

Please find below specific comments:

1) The introduction is to long, must be rewritten to focus on main aspects of the manuscript.

2) Page 3, line 19. The presented paragraph is redundant, it includes description that belong to the method section.

3) Page 3 line 22, wrong method of reference, also this sentence while is related to handstand, press handstand is different gymnastic elements and should not be discussed in this manuscript.

4) Page 4 line 1, wrong reference method, this occurs also in discussion section again.

5) Page 7,9: please write “same investigators” or similar instead of Authors initials.

6) Page 8, The AQV is not clearly described, it is hard to follow, what will be best outcome and what will be the worst (failed to handstand), please indicate what was the maximum of point that participant could achieved due to the errors.

7) Page 9 “arm length times three” it should be rewritten (for example multiplied) as it is misleading.

8) Page 9: I believe that “single arm push-up with legs a pelvic width apart” it is not correct description for this position, as the push-up implies flexing and extending the elbow joints in supporting arm. I recommend something like, single hand front support or similar.

9) Why the quality of handstands was evaluated with two methods? What was the purpose? Each additional testing with similar variable increase potential bias.

10) Lower part of Table 1. Is vague and hard to follow, please change to make it clear.

11) There is no figure 2 introduced in the manuscript.

12) While I understand the purpose of describing entire evaluation process, the details like division into 6 groups, the station approach etc. are non-important for the aim of the study. Please simplify this as it diminishes the clearance of the manuscript.

13) Page 15 lines 1-2, There are missing symbols in the equations.

14) While the Authors recognized the different shoulder position in their testing and handstand (90° vs 180°), more should be elaborated on the anatomy and function of shoulder joint as it comprises of glenohumeral joint and acromioclavicular joint.

6. PLOS authors have the option to publish the peer review history of their article (what does this mean?). If published, this will include your full peer review and any attached files.

Reviewer #1: No

Reviewer #2: No

---

## [Author Response · Author response to Decision Letter 0]

25 Jan 2024

All comments and edits made to the manuscript are attached in a separate file.

---

## [Decision Letter · Decision Letter 1]

14 Feb 2024

PONE-D-23-29129R1Are the shoulder joint function, stability, and mobility tests predictive of handstand execution?PLOS ONE

Dear Dr. Malíř,

Thank you for submitting your manuscript to PLOS ONE. After careful consideration, we feel that it has merit but does not fully meet PLOS ONE’s publication criteria as it currently stands. Therefore, we invite you to submit a revised version of the manuscript that addresses the points raised during the review process.

Both reviewers saw an improvement in the manuscript but raised a number of points that have not be addressed. Please ensure that you have addressed all points raised by the reviewers. 

We look forward to receiving your revised manuscript.

Kind regards,

Aliah Faisal Shaheen

Academic Editor

PLOS ONE

Reviewers' comments:

Reviewer's Responses to Questions

**Comments to the Author**

1. If the authors have adequately addressed your comments raised in a previous round of review and you feel that this manuscript is now acceptable for publication, you may indicate that here to bypass the “Comments to the Author” section, enter your conflict of interest statement in the “Confidential to Editor” section, and submit your "Accept" recommendation.

Reviewer #1: (No Response)

Reviewer #2: (No Response)

2. Is the manuscript technically sound, and do the data support the conclusions?

Reviewer #1: Yes

Reviewer #2: Partly

3. Has the statistical analysis been performed appropriately and rigorously? 

Reviewer #1: Yes

Reviewer #2: Yes

4. Have the authors made all data underlying the findings in their manuscript fully available?

Reviewer #1: Yes

Reviewer #2: Yes

5. Is the manuscript presented in an intelligible fashion and written in standard English?

Reviewer #1: Yes

Reviewer #2: (No Response)

6. Review Comments to the Author

Reviewer #1: Not all of the reviewer's comments have been taken into account, so I am asking for corrections on the following issues:

1. The study still lacks of research questions. Please present research questions in bullet points.

2. Please describe in more detail the data in Table 1.

3. Reference numbers: 22, 24 contain errors. Please proofread and cite these items according to PloS One requirements as follows:

Omorczyk J, Bujas P, Puszczałowska-Lizis E, Biskup L. Balance in handstand and postural stability in standing position in athletes practicing gymnastics. Acta Bioeng Biomech. 2018; 20 (2): 139-147. doi: 10.5277/ABB-01110-2018-02.

Puszczałowska-Lizis E, Omorczyk J. The level of body balance in standing position and handstand in seniors athletes practicing artistic gymnastics. Acta Bioeng Biomech. 2019; 21 (2): 37-44. doi: 10.5277/ABB-01352-2019-02.

When the authors address these issues I will be able to comment definitively and make the final decision.

Reviewer #2: The manuscript has improved since last version, however there are still issues that should be addressed.

Authors must introduce all the abbreviation in full as soon as they appear in main text. Authors have changed some text and now there are abbreviation before the explanations.

1.) Previous 9) “Why was the quality of handstands evaluated with two methods? What was the purpose?

Each additional testing with similar variable increases potential bias.

Thank you for your comment. We chose these two rating scales as they provide different insights into the execution quality. Although the AQV is the official FIG rating scale, it is limited in terms of resolution. While not an official rating scale, the E-score allows for a much finer evaluation. Though these two scales were statistically significantly correlated (as can be expected) in our study, the strength of association was below a level (see page 13, lines 4-5) at which we would consider them interchangeable. Therefore, both provide separate insights into execution quality. We have amended the related sections of the text to reflect better our rationale for using both scales; see page 7, lines 12-24, and page 8, lines 1-10.”

@While Authors improve the rational for the usage of two scores, still it exceeds the purpose of this manuscript. I see and recommend two options: 1) stay with only one score method as It was suggested previously, or 2) incorporate the issue of scoring methods in handstand to the manuscript and thus change the manuscript topic and introduction, aim, discussion etc. accordingly.

2) Page 8. Lines 12-14. This sentence should be earlier not in the description of the stations.

3) Figure 3 and 4. There are no mention about them in the main text, they should also have the abbreviations explained in the figure caption.

4) previous 10) “Lower part of Table 1. Is vague and hard to follow, please change to make it clear.

Thank you for your feedback. We have redesigned Table 1 to make it more straightforward and accessible (see page 12, line 3).”

@Still the table is not so clear to the AQV description and again the abbreviations should be explained in tables.

5) previous 12) “While I understand the purpose of describing entire evaluation process, the details like division into 6 groups, the station approach etc. are non-important for the aim of the study. Please simplify this as it diminishes the clearance of the manuscript.

We agree, and we have thus removed this redundant information from the manuscript. See the rewritten text on page 5, lines 18-21.

@Still there are too much information on the procedure, that is not related directly to the outcome. Now, the Authors write about six groups of 20 so there was 120 participants? Materials describe 111. The indicating of stations it is unnecessary and blurs the image of research.

6) previous 13) Page 15 lines 1-2, There are missing symbols in the equations.

Thank you for noticing. We have added degrees of freedom to the calculation reporting (see page 14, lines 5-6).

@I believe that there should be “ꭓ2” but in PDF version I still see only blank squares.

7) previous 14) While the Authors recognized the different shoulder position in their testing and handstand (90° vs 180°), more should be elaborated on the anatomy and function of shoulder joint as it comprises of glenohumeral joint and acromioclavicular joint.

Thank you for your comment. We have added a new section discussing the key differences in loading of the rotator cuff muscles in different ranges of shoulder flexion; see page 15, lines 19-25, and page 16, lines 1-2).

@While the description improved, in my opinion this part focuses only on the glenohumeral joint and no elaboration on acromioclavicular/sternoclavicular joint is made.

8) Table 2 and 3 should have their captions above the table not underneath.

7. PLOS authors have the option to publish the peer review history of their article (what does this mean?). If published, this will include your full peer review and any attached files.

Reviewer #1: No

Reviewer #2: No

---

## [Author Response · Author response to Decision Letter 1]

25 Mar 2024

All comments and responses are in the attached document "Response to reviewers"

---

## [Decision Letter · Decision Letter 2]

16 Apr 2024

Are the shoulder joint function, stability, and mobility tests predictive of handstand execution?

PONE-D-23-29129R2

Dear Dr. Malíř,

We’re pleased to inform you that your manuscript has been judged scientifically suitable for publication and will be formally accepted for publication once it meets all outstanding technical requirements.

Kind regards,

Aliah Faisal Shaheen

Academic Editor

PLOS ONE

Additional Editor Comments (optional):

Reviewers' comments:

Reviewer's Responses to Questions

**Comments to the Author**

1. If the authors have adequately addressed your comments raised in a previous round of review and you feel that this manuscript is now acceptable for publication, you may indicate that here to bypass the “Comments to the Author” section, enter your conflict of interest statement in the “Confidential to Editor” section, and submit your "Accept" recommendation.

Reviewer #1: All comments have been addressed

2. Is the manuscript technically sound, and do the data support the conclusions?

Reviewer #1: Yes

3. Has the statistical analysis been performed appropriately and rigorously? 

Reviewer #1: Yes

4. Have the authors made all data underlying the findings in their manuscript fully available?

Reviewer #1: Yes

5. Is the manuscript presented in an intelligible fashion and written in standard English?

Reviewer #1: Yes

6. Review Comments to the Author

Reviewer #1: (No Response)

7. PLOS authors have the option to publish the peer review history of their article (what does this mean?). If published, this will include your full peer review and any attached files.

Reviewer #1: No
